# Deep brain stimulation of the internal capsule enhances human cognitive control and prefrontal cortex function

A.S. Widge[1,2,5], S. Zorowitz [1,6], I. Basu[1], A.C. Paulk [3], S.S. Cash[3], E.N. Eskandar[4,7], T. Deckersbach[1], E.K. Miller[2] & D.D. Dougherty[1]

Deep brain stimulation (DBS) is a circuit-oriented treatment for mental disorders. Unfortunately, even well-conducted psychiatric DBS clinical trials have yielded inconsistent symptom relief, in part because DBS' mechanism(s) of action are unclear. One clue to those mechanisms may lie in the efficacy of ventral internal capsule/ventral striatum (VCVS) DBS in both major depression (MDD) and obsessive-compulsive disorder (OCD). MDD and OCD both involve deficits in cognitive control. Cognitive control depends on prefrontal cortex (PFC) regions that project into the VCVS. Here, we show that VCVS DBS' effect is explained in part by enhancement of PFC-driven cognitive control. DBS improves human subjects' performance on a cognitive control task and increases theta (5–8Hz) oscillations in both medial and lateral PFC. The theta increase predicts subjects' clinical outcomes. Our results suggest a possible mechanistic approach to DBS therapy, based on tuning stimulation to optimize these neurophysiologic phenomena.

---

[1] Department of Psychiatry, Massachusetts General Hospital and Harvard Medical School, 149 13th St, Boston, MA 02129, USA. [2] Picower Institute for Learning and Memory, Massachusetts Institute of Technology, 43 Vassar St, Cambridge, MA 02139, USA. [3] Department of Neurology, Massachusetts General Hospital and Harvard Medical School, 55 Fruit St, Boston, MA 02114, USA. [4] Department of Neurological Surgery, Massachusetts General Hospital and Harvard Medical School, 55 Fruit St, Boston, MA 02114, USA. [5] Present address: University of Minnesota, 2001 6th St SE, Minneapolis, MN 55455, USA. [6] Present address: Princeton Neuroscience Institute, Princeton, NJ 08540, USA. [7] Present address: Department of Neurological Surgery, Albert Einstein College of Medicine—Montefiore Medical Center, Bronx, NY 10467, USA. Correspondence and requests for materials should be addressed to A.S. W. (email: awidge@umn.edu)

Mental disorders, particularly mood and anxiety disorders, are a leading cause of disability and economic burden[1]. This is in part because many patients have no relief from gold standard clinical treatments. Focused electrical/magnetic brain stimulation has been proposed as a better approach to the mental health epidemic, because stimulation therapies may directly affect the circuit deficits believed to underlie mental illness[2]. Deep brain stimulation (DBS) is a particularly promising new therapy. Early DBS studies in major depressive disorder (MDD) and obsessive–compulsive disorder (OCD) were extremely encouraging[3,4]. Patients reported dramatic and long-lasting symptom relief where all prior treatments had failed. Of five blinded and sham-controlled DBS clinical trials, however, two met their primary endpoint, two did not, and one remains unpublished[3–5]. This ambiguous set of outcomes has limited DBS' use in mental illness, despite the pressing need for new treatments for these disorders. We and others have argued that the limited clinical trial signal does not reflect a lack of efficacy, but instead a limited mechanistic understanding[4–6]. DBS therapy requires fine tuning of individual patients' stimulation parameters, altering the applied electric field to engage a target circuit[7]. A challenge in mental illness is that there is no objective biomarker of that engagement, and thus no rigorous definition of "effective dose"[8,9]. The brain's response to electrical stimulation has been studied for decades, from simple preparations to complex modeling[7,10,11]. Despite those studies, the precise therapeutic mechanism of DBS' high-frequency stimulation is a topic of frequent debate. Current theories focus around resetting of abnormal oscillatory activity, which may alter information transmission in distributed circuits[8,12]. This uncertainty suggests that some patients who did not respond in clinical trials likely did not receive an active dose of the study intervention. Others may not even have had a circuit deficit that was appropriate for DBS treatment. This stands in contrast to the testing of more common therapies such as medications, where investigators can at least be certain that a drug achieved a desired serum level.

In clinically successful DBS applications, such as Parkinson disease, the dosing problem can be solved by trial and error. Motor symptoms change almost immediately when stimulation is optimal, and hence clinicians can find the correct dose in a matter of hours. Psychiatric DBS changes symptoms over weeks to months, making it impossible to fully explore the parameter space or to immediately verify the appropriateness of a dose adjustment. If DBS' mechanisms were better understood, it might be possible to redesign the clinical approach around physiology. Stimulation could be titrated to achieve a specific and relatively immediate electrophysiologic change, with symptom relief emerging in response to that change[4,8,13,14]. Thus, understanding mechanisms of action at the neurophysiologic level is a critical next step for developing DBS as a psychiatric treatment.

DBS of one specific brain region, the ventral internal capsule/ventral striatum (VCVS), may be helpful in identifying some of those therapeutic mechanisms. VCVS is the only DBS target to pass blinded trials, with success in both MDD and OCD[3,4]. These disorders are clinically considered as very different, but their common response to VCVS stimulation suggests the presence of common underlying pathophysiology. One commonality particularly relevant to DBS is that MDD and OCD both involve impaired cognitive control[5,15]. Cognitive control is the flexible adjustment of mental processing and/or responses in the face of changing environmental demands[16,17]. Control deficits may explain inflexible behavior in many mental disorders, e.g., the "automatic" negative cognitions of MDD, the repetitive behaviors of OCD, or the rigid interests of autism[15,18]. The converse (flexibility) is critical to clinical recovery, e.g., when a behavior therapist asks a patient to act opposite to a habit. Brain structures involved in cognitive control include medial prefrontal cortex (mPFC), the dorsal anterior cingulate (dACC)[16,18], lateral prefrontal cortex (PFC)[18,19], and recurrent circuits connecting those structures to striatum[20]. Those cortico-striatal circuits pass through the VCVS DBS target, meaning that stimulation should broadly influence prefrontal networks[21,22]. Thus, we propose that VCVS DBS may act in part by enhancing cognitive control, possibly by retrograde activation of corticofugal fibers in the ventral capsule.

Experimentally, control is often studied through conflict tasks, where subjects must suppress a pre-potent response to follow a less intuitive rule[16,17]. When performing conflict tasks, humans and other species show increased low-frequency oscillations of the electrical local field potential (LFP) and/or electroencephalogram (EEG). These are particularly common in the theta (4–8 Hz) band and over/within mPFC[17,23]. Theta

### Table 1 Subject characteristics

| Diagnosis | Age/sex | YBOCS BL | YBOCS FU | MADRS BL | MADRS FU | Responder | Task EEG | Rest EEG | Stim Freq |
|---|---|---|---|---|---|---|---|---|---|
| OCD | 30/F | 34 | 12 | 34 | 11 | Y | Y | N | 130 |
| OCD | 30/F | 31 | 27 | 2 | 3 | Y | Y | Y | 130 |
| MDD | 40/F | N/A | N/A | 44 | 11 | Y | N | N | 130 |
| MDD | 30/F | N/A | N/A | 35 | 28 | N | N | N | 130 |
| MDD | 60/F | N/A | N/A | 44 | 4 | Y | N | N | 130 |
| MDD | 50/M | N/A | N/A | 33 | 10 | Y | Y | Y | 130 |
| MDD | 60/M | N/A | N/A | 33 | 10 | N | Y | N | 100 |
| MDD | 50/F | N/A | N/A | 42 | 29 | N | Y | N | 90 |
| MDD | 60/M | N/A | N/A | 36 | 10 | N | Y | N | 130 |
| MDD | 50/F | N/A | N/A | 42 | 28 | Y | N | Y | 130 |
| MDD | 50/M | N/A | N/A | 38 | 24 | N | Y | Y | 50 |
| MDD | 60/M | N/A | N/A | 34 | 9 | Y | Y | Y | 120 |
| MDD | 70/F | N/A | N/A | 38 | 30 | N | N | N | 130 |
| MDD | 50/M | N/A | N/A | 35 | 38 | N | N | N | 100 |

Ages have been rounded to the nearest decade to mask identities. "Diagnosis" refers to the primary indication for receiving VCVS DBS. YBOCS scores were not collected for subjects whose primary clinical complaint was not OCD. The "Responder" column indicates whether the subject achieved clinical response at any point during his/her clinical trial; the criterion was a 50% drop in MADRS or a 35% drop in YBOCS. This may not have corresponded to the score at the time of study data collection. Response prediction analyses were based on the score at the time of recording for this study, as this was more likely to correlate to the measured biomarkers. The "Task EEG" and "Rest EEG" columns indicate subjects who contributed technically adequate EEG for MSIT-related and resting-state analysis, respectively. All subjects underwent recordings, but some could not be analyzed due to excessive artifact, most commonly from vocalization or substantial head/face muscle activation. "Stim Freq" gives the frequency of DBS, in Hz

MDD major depressive disorder, OCD obsessive–compulsive disorder, YBOCS Yale-Brown obsessive–compulsive scale, MADRS Montgomery–Asberg depression rating scale, BL baseline (just before implant), FU follow-up (date of EEG recording or nearest clinical visit)

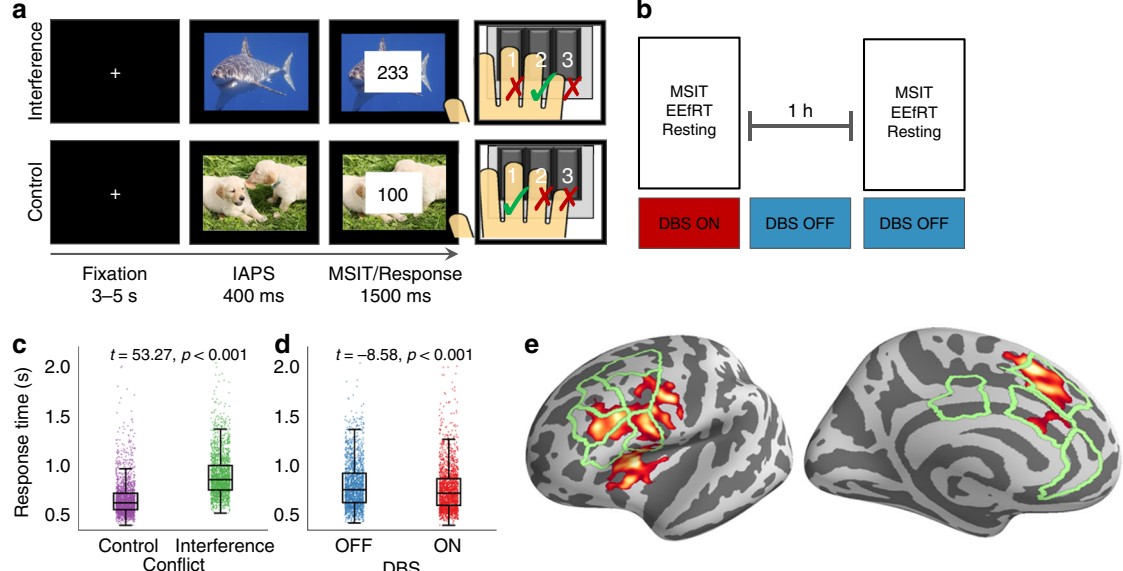

**Fig. 1** Experiment design and behavior. **a** Affective multi-source interference task (MSIT). Subjects chose the number that differed from its neighbors while ignoring an emotionally arousing image. On interference trials (top), a non-intuitive motor mapping and flanking distractors increased cognitive load. **b** Subjects performed task and resting recordings with deep brain stimulation (DBS) ON and again after 60 + minutes with DBS OFF. **c** Interference increased response times (RTs) ($n = 3785$ trials from 14 subjects, $t = 53.3$, $p < 1e\text{-}20$, Wald test of generalized linear model (GLM) coefficient). **d** DBS reduced RTs by 34 ms ($t = -8.6$, $p < 1.33e\text{-}17$). Error bars represent standard error of the mean. **e** Atlas labels for source localization overlaid on functional MRI interference/control contrast from past studies

oscillations are theorized to allow mPFC neural ensembles to synchronize with and drive neural firing in other brain regions, allowing mPFC to "gate" response sets and behavior styles[24]. Thus, if DBS does enhance cognitive control, the enhancement should be reflected in more powerful PFC theta oscillations. This would also comport with broader theories that DBS acts by restoring healthy oscillatory activity. We tested this hypothesis by manipulating human subjects' VCVS DBS and measuring both medial and lateral PFC activity via EEG as they performed a difficult cognitive control/conflict task. Here, we show that VCVS DBS enhances cognitive control, that this enhanced control is correlated with the expected PFC theta oscillations, and that the increased theta power is in turn correlated with clinical recovery. It should thus be possible to improve DBS' clinical efficacy by using these markers for biomarker-based, closed-loop stimulator programming.

## Results

**VCVS DBS enhances cognitive control.** Fourteen subjects (12 MDD and 2 OCD; Table 1) with VCVS DBS implants performed a variant of the Multi-Source Interference Task (MSIT, Fig. 1a) that included emotional distractors to increase overall cognitive load. We recorded EEG as subjects performed MSIT with their usual clinical stimulation either ON or OFF (Fig. 1b). We analyzed task response times (RTs) in a mixed effects generalized linear model (GLM). Cognitive conflict (interference), emotional distraction, and DBS all influenced RT (Fig. 1c, d, Supplementary Fig. 1). As in prior studies[25,26], interference slowed RTs by 224 ms on average ($t = 53.2$ for Wald test of GLM coefficient, $p <$ 1e–20, Fig. 1c). DBS enhanced performance: subjects were on average 34 ms faster with DBS ON ($t = -8.6$, $p < 1.33e\text{-}17$, Fig. 1d). There was no interaction between DBS and trial type, i.e., the RT for both control and interference trials was reduced (Supplementary Fig. 1). DBS-related improvement was not explained by a speed-accuracy tradeoff, as there was no difference in error rate between ON and OFF (1.74 vs. 1.69% respectively, $p = 0.48$, binomial test). It was not explained by psychomotor

changes, as a different task performed minutes later (Fig. 2a) showed no change in button pressing speed with DBS ON vs. OFF (Fig. 2b). On that second task, DBS also slowed reaction times as subjects chose between two rewarding options, suggesting that the MSIT improvement is not driven by impulsive responding (Fig. 2c, d). The effect also is not explained by practice on the task. Any practice effect in this ON-then-OFF design would appear as the exact opposite of our observation (faster in OFF). Further, a cohort of subjects who performed repeated blocks of MSIT without DBS manipulations showed no change in RT from block to block (Supplementary Fig. 2).

**DBS' effects on cognitive control are linked to PFC theta oscillations.** To test the relationship between improved cognitive control and frontal theta oscillations, we source localized the task-related EEG activity to cortical regions implicated in cognitive control and MSIT specifically (Fig. 1e). We verified that the source localization preserved oscillatory activity and that the majority of the task-related activity was non-phase-locked (Supplementary Fig. 2). We then tested for significant theta modulation through sliding multivariate regression with temporal cluster correction (see Methods). As expected, the cognitive effort required to perform MSIT increased the power of non-phase-locked theta oscillations throughout PFC (Fig. 3a, b). In ventrolateral PFC (anterior inferior frontal gyrus) particularly, theta power increased over baseline for the entire post-MSIT period (Fig. 3c, d). DBS potentiated that increase ($p < 0.05$, cluster mass corrected via permutation testing) for nearly the entire MSIT decision-making period (Fig. 3c) and −199 ms to +120 ms around the response (Fig. 3d). The DBS effect was specific to the theta band, with few changes in other frequencies (Fig. 3e, f, Supplementary Figs. 3–4). DBS' effect on theta was specific to the active exercise of control for decision making—theta power in resting-state recordings from the same subjects did not change between DBS ON and OFF conditions (Supplementary Figure 4). In the task recordings, 79.93% of the stimulus-locked significant theta cluster mass (summed across labels) was between the MSIT

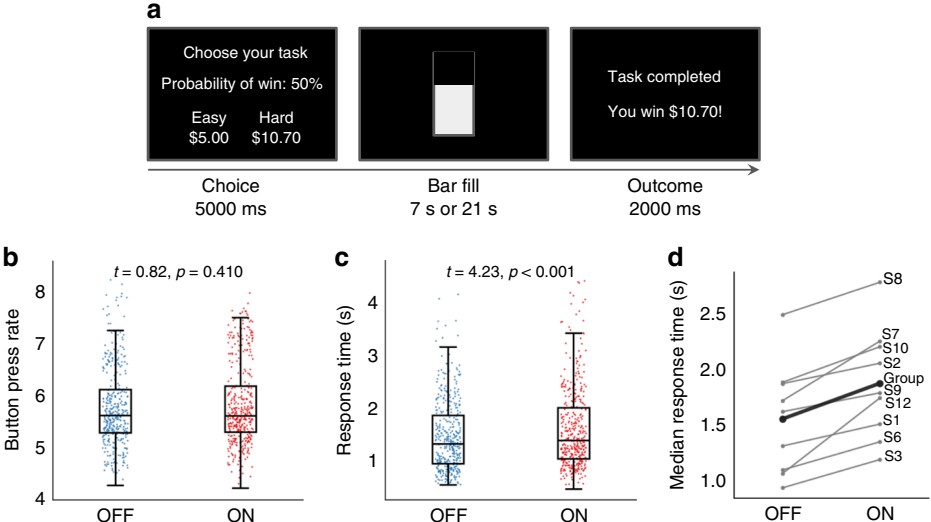

**Fig. 2** Deep brain stimulation (DBS') beneficial effects are specific to conflict decision-making. **a** Effort expenditure for rewards task (EEfRT)[27]. Subjects are presented with a choice of an "easy" or "hard" trial. In either, they must press a key quickly many times to fill up a bar, but "hard" trials require more presses and use the non-dominant hand. "Hard" trials offer a larger but uncertain reward, forcing subjects to quickly make a choice between gains. Choices are presented with time pressure that prevents formal expected value calculations. The subject then performs their chosen key-pressing within a set interval (7 s for easy, 21 s for hard), with a chance of reward (right panel). Failure to perform sufficient key presses in time yields no reward. **b** DBS ON/OFF difference in button pressing rates during EEfRT. Press rates under the two conditions were not significantly different (mixed effects Gamma regression with nine participants and 794 observations; $t = 0.824$, $p = 0.410$), suggesting that ventral internal capsule/ventral striatum (VCVS) DBS has no effect on movement speed. This also argues against a fatigue effect. Plotting conventions follow Fig. 1. **c** DBS ON/OFF difference in choice response times (RTs) during EEfRT. DBS increased these RTs, with a median difference of 66.1 ms (mixed effects Gamma regression with nine participants and 809 observations; $t = 4.231$, $p < 0.0001$). This is opposite to MSIT, despite the EEfRT being performed only a few minutes later. This argues that the cognitive control effect is specific to the functions tested by MSIT. It also argues that the DBS ON < OFF RTs in MSIT are not explained by order/fatigue effects, which would produce ON < OFF during EEfRT also. **d** DBS ON/OFF difference in choice RTs, as in (**c**), for individual subjects. All subjects showed ON > OFF choice RTs in EEfRT, which likely represents increased deliberation. This argues against DBS-induced impulsivity as an explanation of the main finding. Both impulsive choice and fatigue would produce an OFF > ON pattern. Labels follow Table 1

onset and the mean RT (Fig. 3g), whereas 87.73% of the significant response-locked mass was before the response.

In time-domain evoked potentials, almost no DBS effect survived multiple-testing correction (Fig. 4; Supplementary Fig. 5). The DBS effect was separate in location and frequency from task-induced Interference effects (Fig. 3g vs. Fig. 4g, Supplementary Fig. 5). Supporting the claim that theta power is mechanistically liked to cognitive control, ON–OFF theta power changes at the single subject level correlated with subjects' RT improvement (Supplementary Fig. 6). These results follow multiple prior studies suggesting that theta oscillations in medial PFC, lateral PFC, and cingulate are a neurophysiologic correlate of effective cognitive control. They further suggest that amplifying those oscillations facilitates control.

**Augmented conflict-evoked PFC theta is a biomarker of response to DBS**. These neurophysiologic changes predicted clinical outcomes. ON–OFF RT changes alone did not correlate with subjects' clinical response to DBS (Fig. 5a, b). However, theta change in anterior inferior frontal gyrus (IFG), the area with the largest task and DBS theta effects, did correlate. The ON–OFF theta difference of Fig. 3c, as calculated in individual subjects, significantly correlated with improvement in depressive symptoms from pre-operative baseline ($n = 8$, $r = 0.76$, $p = 0.03$ by Fisher Z-transform; area under curve 0.87 with confidence interval (CI) 0.57–1.0; Fig. 5c, d). This correlation appeared to be true regardless of subjects' initial clinical diagnosis. Neither RT nor theta EEG was strongly associated with hypomania (Fig. 5e–h), a major clinical complication of VCVS DBS[28]. Hypomania is one of several manifestations of DBS-induced impulsivity. Its lack of correlation with theta or RT, combined

with the lack of impulsive choice in a companion task involving win-win comparisons (Fig. 2), suggests that our results cannot be explained simply by impulsive responding.

## Discussion

A popular theory of DBS' mechanism of action suggests that this deep brain intervention acts primarily on cortex, by stimulating cortical projections to the implant site[21,29]. Our results support this theory, finding oscillatory changes in multiple regions whose thalamic projections traverse the VCVS[21]. The effect is only seen in task-related theta, suggesting that DBS specifically modulates functional ensembles that activate during effortful cognitive control. The correlation between that theta modulation and clinical outcomes suggests a potential physiologically informed approach to psychiatric DBS. The present practice of adjusting stimulation based on patients' subjective report of immediate mood changes leads to clinician/patient frustration, adverse clinical outcomes, and missed clinical trial endpoints[5,13,30,31]. In future studies, stimulation parameters might directly be titrated to change a theta biomarker, rather than relying on ad hoc clinician opinion. For instance, patients might continuously perform tasks requiring effortful cognitive control, with continuous monitoring of the trial-to-trial induced oscillations. Stimulation parameters such as intensity and location along the dorso-ventral axis of the internal capsule could then be adjusted, either manually by a programming clinician or under computer control. Next-generation DBS hardware is already capable of self-titrating stimulation to achieve an electrophysiologic response[32], and we have described early examples of frameworks for this type of task-driven programming[10,13,33]. We have also demonstrated methods for frequency-specific oscillatory enhancement[34]. Combining

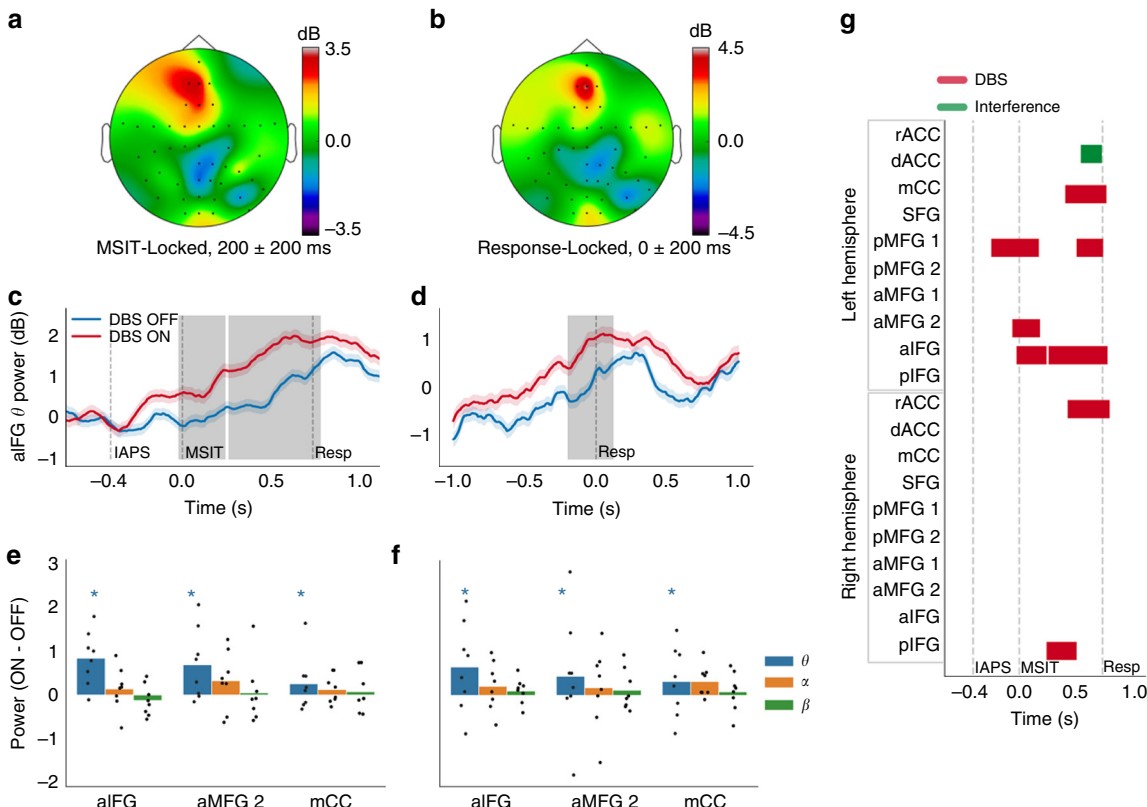

**Fig. 3** Deep brain stimulation (DBS) increases task-evoked frontal theta oscillations. **a**, **b** Topographic plots of **a** stimulus- and **b** response-locked theta power. **c**, **d** Theta time courses in left anterior inferior frontal gyrus (aIFG), locked to **c** stimulus presentation or **d** response. Shading is standard error of the mean from 1000 bootstraps. Gray bars indicate significant cluster masses from sliding regression at $p < 0.05$, corrected via permutation testing. **e**, **f** DBS ON-OFF theta (4–8 Hz), alpha (8–15 Hz), and beta (15–30 Hz) change during the **a**, **b** time windows, in three example labels, (**e**) stimulus and **f** response locked. Error bars denote standard error of the mean. Stars denote the presence of a $p < 0.05$ cluster as in **c**, **d**, which may not necessarily be during this illustrative time window. Thus, significance in this plot does not directly correlate with the means and error bars, but correlates with the more rigorous statistics of **c**, **d** and **g**. Individual points reflect change in power for each patient. **g** Extents of significant DBS and interference clusters. a/A anterior, d dorsal, I inferior, m/M medial, r rostral, S superior, CC cingulate cortex, FG frontal gyrus

these methods would bring psychiatric neurostimulation closer to movement disorders, where tremor can be quickly observed and DBS titrated to suppress it[35].

Prospective validation is, however, critical before moving to a closed-loop trial. This was a small study, despite being an exhaustive sample of all willing subjects at one research center. We used robust procedures to prevent model over-fitting, and information criterion minimization in particular is mathematically equivalent to the gold standard, out-of-sample cross-validation[36]. The heterogeneity of psychiatric illness nevertheless makes validation an important step for any putative biomarker[37,38]. Ideally, the procedures performed here should be administered pre-operatively and sequentially during a patient's DBS treatment, demonstrating a longitudinal correlation between task-induced theta and clinical response. An ongoing study (NCT03184454) aims at that prospective longitudinal validation. Further, clinical response correlated with changes in task-induced theta, but did not correlate with DBS-induced behavior change—even though behavior itself was correlated to theta power, as it was in other cognitive control studies[17,23,39]. This may reflect more on the task than on the construct—we chose the MSIT as our index because of its ability to generate subject-level significant effects. The tradeoff was a task that could be performed with few errors. In another recent study with a cognitive control task involving a higher error rate, behavior did track clinical improvement[39]. Measuring control with multiple tasks would be

an important consideration for future work. A more nuanced measure of cognitive control might also compensate for the heterogeneity and non-interval nature of clinical scales such as the MADRS.

Given the broad role of cognitive control deficits in mental illnesses, our results could have equally broad clinical applications. Other DBS targets for psychiatric illness, such as the subcallosal cingulate (SCC) and the medial forebrain bundle, also project to the regions we studied[40,41]. Augmented cognitive control might be a common mechanism of DBS at multiple clinical targets. Alternatively, the slightly different white matter projections of each DBS site might allow each to access and improve a different cognitive function. In contrast, a DBS target for movement disorders and OCD, the subthalamic nucleus (STN), tends to increase impulsive responding during conflict[42,43]. STN-like impulsivity thus does not appear to explain our results. We did not observe a specific speedup on high-conflict trials, which was a hallmark of the STN studies. Further, in a second task run immediately after MSIT, subjects had significantly increased RTs when required to deliberate in a win–win situation. This is the opposite of the STN result. Combining the effects of these different targets might enable fine-grained control of individual patients' cognitive/emotional deficits, a "precision medicine" approach to therapy[5,13]. In support of that idea, a recent clinical trial combined VCVS and STN stimulation in OCD patients. When STN stimulation modulated

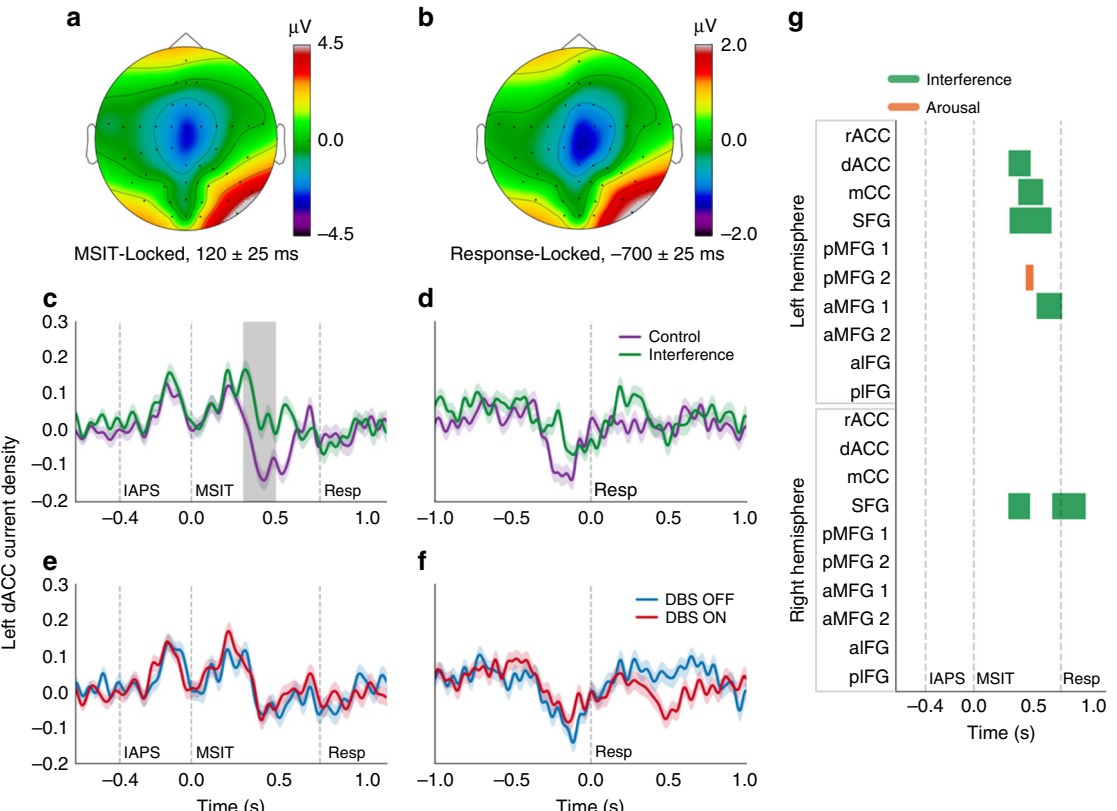

**Fig. 4** Interference, but not deep brain stimulation (DBS), modulates time-domain evoked potentials (ERPs) in source space. **a, b** Grand mean topographic plots of **a** stimulus- and (**b**) response-locked ERP. Voltages in control and interference trials were normalized against their respective pre-trial baselines before averaging. **c**–**f** Source-localized time courses of current density attributable to the left dorsal anterior cingulate gyrus (dACC) label, shown locked either to MSIT presentation (**c, e**) or response (**d, f**), and for both the control/interference contrast (**c, d**) and DBS ON/OFF contrast (**e, f**). dACC is often identified as the putative generator of midline frontal negativity in conflict tasks. "IAPS", "MSIT", and "Resp" denote the IAPS image onset, MSIT number stimulus onset, and mean response time (RT), respectively. Shading around each line is standard error of the mean computed from 1000 bootstrap resamples. Gray bar in **c** indicates a significant cluster mass at $p < 0.05$, corrected via permutation testing, signifying a difference between conditions (based on coefficients of sliding multivariate regression) from 291 to 473 ms. This is the only cluster in this label that survived FDR correction for multi-label testing; no DBS effect survived. **g** Time course of significant effects in the stimulus-locked analysis, corresponding to gray shading in **c**. Interference and arousal were the only factors that significantly influenced ERP amplitude in any region, with Interference effects throughout the dorsal frontal midline. a/A anterior, d dorsal, I inferior, m/M medial, r rostral, S superior, CC cingulate cortex, FG frontal gyrus

corticothalamic fibers originating in PFC, patients performed better on a Stroop-like task[44]. These are the same fibers we are likely to have engaged using our larger VCVS leads.

Theta oscillations and metrics derived from theta have previously been proposed as biomarkers of depressive states and treatment response[38,45]. One article proposed another theta measure, resting-state prefrontal cordance, as a predictor of SCC DBS response[46]. However, as we reviewed in a recent meta-analysis[38], those studies focused on resting-state activity, where a given oscillatory band may represent any number of ongoing mental processes. Here, we specifically demonstrated a change in task-induced frontal theta, a neurophysiologic process with strong and well-documented links to executive function[17,23]. The report of resting-state cordance in SCC DBS also did not reach its primary statistical significance endpoint and did not demonstrate out-of-sample reliability. We reached our pre-specified significance level even after multiple-comparison correction, and we demonstrated out-of-sample evidence through bootstrapping and through information criterion minimization (mathematically equivalent to cross-validation[36]). We posit that the use of a specific cognitive task to enhance the theta output of conflict-related ensembles greatly improved signal-to-noise, aiding detection. We also note that we observed performance improvements (RT decreases) on both control and interference

trials. We attribute this to the task structure, in which trial types were interleaved, both with high frequency, and in an unpredictable fashion. In this design, subjects cannot prospectively predict when they will need to exercise cognitive control. This should cause control-related frontal circuits to remain in a state of readiness, effectively decreasing the burden of "switching on" control[16].

We found enhanced theta oscillations in medial and lateral PFC, dorsally and ventrally, but cognitive control might depend on only a subset of these regions. That possibility could be explored in animal models, where genetic tools could limit DBS' effects to a specific PFC projection[29]. It would be useful to explore the degree to which this effect requires specific neurotransmitters within those circuits. DBS of this same region was recently suggested to affect metabolism specifically through nucleus accumbens D1-receptor cells, and those same dopaminergic circuits may be relevant for DBS' psychiatric effects[47]. Finally, animal studies might dissect the acute vs. chronic effects of neurostimulation. In our data, a greater DBS-induced theta change was associated with a smaller clinical effect. This may represent a neuroplastic effect of chronic stimulation: patients who experience a strong "theta rebound" with DBS OFF may not have experienced permanent remodeling of brain networks, and thus retain their depressive vulnerability. Clarifying these mechanisms

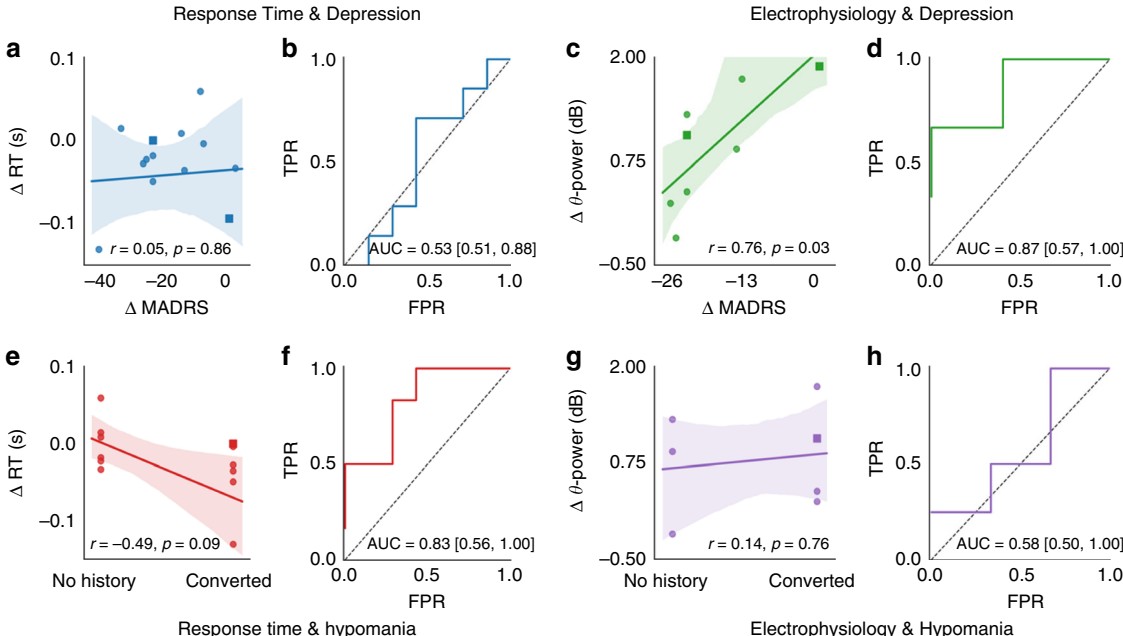

**Fig. 5** Changes in ventrolateral PFC (VLPFC) theta power predict clinical outcomes. **a, c, e, g** Clinical outcomes vs. behavioral and neural changes. Panels (**a, c**) show change in Montgomery–Åsberg Depression Rating Scale (MADRS, where negative values indicate improvement); **e, g** show hypomania. *Y* axes are **a, e** the DBS-induced response time (RT) difference for each subject or **c, g** theta power difference during the period covered by the Fig. 3c cluster. Lines represent a robust linear regression. Circles represent subjects with primary diagnosis of MDD, squares represent subjects with primary diagnosis of OCD. Shaded areas are confidence bounds from 1000 bootstraps. "r" represents Pearson correlation, *p*-value via Fisher *Z*-transformation. **b, d, f, h** Receiver operating characteristic (ROC) curves for clinical response prediction. Confidence interval for area under the curve (AUC) is from 1000 bootstrap draws

and the clinical correlation could advance both the reliability of clinical neuromodulation and our broader understanding of human cognition.

## Methods

**Experimental design**. The overall objective of the study was to assess whether VCVS DBS modulated human cognitive control capabilities and cortical neural oscillations relevant to those capabilities. This was a within-subjects design, where individual subjects completed identical measurement protocols with stimulation on and off. This design increased statistical power (compared with alternate designs where DBS subjects would be compared against non-implanted controls) by admitting hierarchical/mixed modeling and controlling for the substantial heterogeneity of treatment-resistant psychiatric patients. Our pre-specified hypotheses were that:

1. DBS would improve human cognitive control, reflected in increased performance in the DBS ON condition.
2. DBS would augment the power of theta oscillations, primarily in lateral prefrontal cortex and dorsal anterior cingulate cortex, given the specific role of these oscillations in decision-making and response inhibition.
3. The degree of DBS-induced change in the above propositions would explain part of its mechanism of action, as determined by prediction of the clinical outcome.

These analyses were not pre-registered. At the time of data collection and study conception, pre-registration was not a widely available service.

**Subjects**. Fourteen subjects with VCVS DBS consented to participate in the experiments. All had received VCVS DBS implants for a prior clinical study (NCT00640133, NCT00837486, or NCT00555698), with entry criteria given in[48,49]. All were right handed. The sample included six males and eight females, aged in their 30s–70s at time of data collection, with a minimum of 6 months' exposure to chronic stimulation and a maximum of 7 years. Subjects had predominantly been implanted for MDD, but 2/14 had a primary indication of OCD with comorbid MDD. Most had at least partial clinical response to DBS. Informed consent for study participation was obtained by a physician who was not the subject's primary DBS clinician, after the full nature and possible consequences of the study were explained. All study procedures comport with applicable governmental and institutional ethical guidelines. Study procedures were reviewed and approved by the Massachusetts General Hospital Institutional Review Board.

**Experimental protocol**. To probe cognitive flexibility, we employed a modified version of the MSIT (main text Fig. 1a). The MSIT requires subjects to identify which of a set of three numbers is different than its neighbors. Subjects must keep three fingers of their right hand positioned over response keys corresponding to the digits 1–3. In control (non-interference) trials, the target is in the same spatial position as its corresponding response key, and the flanking digits are not valid responses (i.e., they are 0s). In interference trials, the target is out-of-position relative to its corresponding key-press and is flanked by other viable targets. MSIT has been shown to produce robust functional magnetic resonance imaging (fMRI)[25] and electrophysiologic[26] changes, with a significant (interference–control) difference often detectable at the individual subject level. We note that this specific operationalization of cognitive control, performance on a conflict task, is only one of many possible experimental approaches. Cognitive control is evoked in many situations, including approach-avoidance conflict[50], switch-stay decisions[16,51], and possibly also in emotionally valenced self-regulation[52]. The specific advantage of MSIT is that it is verified to induce statistically robust subject-level effects, at both the behavioral and neural level, amplifying our power to detect DBS-induced differences. We further added an emotional interference dimension, based on a hypothesis that subjects with severe treatment-resistant illness would be attentionally biased towards negative pictures. Before each MSIT trial, an image selected from the International Affective Picture System, or IAPS[53], was presented. The image remained on-screen, partially obscured by the MSIT stimulus, for the trial duration. A fixed subset of 144 images were selected from the overall IAPS dataset to cover the range of available valence (positive, neutral, and negative) and emotional arousal ratings.

Each block of trials contained 72 control and 72 interference trials. We assigned positive, neutral, and negative IAPS images assigned to each trial type in a counterbalanced fashion, such that each image was presented once in a control and once in an interference context. The 144 images were split between these two 144-trial blocks in a manner that minimized the mean squared pairwise differences between image ratings when rank ordered by their valence. To prevent response sets or habituation, trial sequence in each block was pseudo-randomized so that subjects never had more than two trials in a row that shared the same valence, interference level, or desired response finger. This highly interleaved trial design was expected to place greater demands on cognitive control systems by reducing predictability of the stimuli. As shown in Fig. 1a, subjects viewed the IAPS picture alone for 400 ms, were presented with the MSIT stimulus and given up to 1500 ms to respond, and then viewed a fixation cross for 3–5 s (randomized with a uniform distribution). They were instructed to minimize eye blinking during the trial and to blink freely during the fixation period. Before data collection, subjects performed a block of 20 trials where they received correct/incorrect feedback, followed by another block of 40 trials without feedback. They repeated this practice, if necessary, until they achieved over 90% correct responses (counting missed trials as incorrect).

Many of our subjects had prior negative life experiences with specific associations to themes presented in IAPS. To control for these strong subjective/idiosyncratic interpretations in this small sample, we collected individual image ratings. After each block was complete, subjects were re-presented with each IAPS image and given 25 s to rate the image emotionally. We used the same self-assessment manikin system originally used to develop the IAPS[54], which assigns each picture a valence rating from 1 to 9 (representing most negative to most positive) and an arousal rating from 1 to 9 (representing not-at-all arousing to highly arousing). Both the MSIT and the post-task IAPS rating images were presented using Psychophysics Toolbox (http://psychtoolbox.org) running under MATLAB 2013a.

Electroencephalographic data were acquired at 1450 Hz (Nexstim eXimia EEG) from 60 channels placed according to the international 10–20 system and the manufacturer's standard cap. The ground electrode was placed on the bridge of the nose. One diagonal bipolar electro-oculogram channel was placed around the right eye. Channels were prepared to <5 kΩ impedance. The scalp location of each channel was digitized after cap preparation and prior to recordings. We also digitized the nasion and both pre-auricular points, plus 100 additional scalp points not corresponding to any EEG sensor, to improve the quality of MRI-to-digitization co-registration. In four subjects, in addition to the task data, we collected 1 min each of eyes-open and eyes-closed resting data just after each task block and before the IAPS self-assessment ratings.

All subjects first completed an MSIT block, resting-state collection, and image assessment with their DBS on at its usual clinical settings (DBS ON). Directly after MSIT, but before resting-state and image-rating blocks, subjects also completed 15 min of the Effort Expenditure for Rewards Task (EEfRT)[27]. A trained clinician then de-activated the bilateral implanted neurostimulators, and the subject rested for at least 1 h without removing the EEG cap. In animal studies, an hour's withdrawal of chronic stimulation was sufficient to produce robust changes in neural activity that appeared to be a rebound/counter-regulatory response[55]. This rebound effect does not terminate within an hour, but persists for an extended period, as documented by clinical studies where patients slowly relapse over a week after DBS discontinuation[56]. The presence of this rebound effect should emphasize or amplify the neurological changes caused by chronic stimulation. After re-preparing any high-impedance channels, subjects again performed MSIT, EEfRT, resting-state, and image ratings (DBS OFF condition) before neurostimulator re-activation. Subjects were aware of their device status, as were the experimenters, although no subject experienced adverse psychological consequences from the study manipulation.

**EEG preprocessing**. EEG analyses used the minimum norm estimate (MNE)-Python suite[57]. Offline, EEG data were bandpass filtered between 0.5 and 50 Hz, then epoched. This effectively removes the DBS artifact as shown in our and others' past work[37,58], as all subjects' stimulators were set above the cutoff frequency. Harmonic frequencies of DBS stimulation would similarly be entirely outside the passband of this filter and outside of all frequency bands analyzed in this work. See Table 1 for individual subjects' stimulation frequencies. We removed eyeblinks and muscle artifacts with signal space projection[59]. We then cut trials/epochs from the continuous data. Stimulus-locked analyses used data from 1.5 s before the IAPS onset to 3.4 s after IAPS onset (1500 ms after end of trial). Response-locked analyses used −1.5 s before to 1.5 s after the response. Amplitude rejection (threshold = ± 150 μV) removed trials with residual artifacts. Finally, we converted all trials to change relative to baseline, defined as 0.5 s to 0.1 s before the IAPS onset. For time-domain analyses, we subtracted the mean of this window from all trials for that specific subject; for frequency-domain, we converted data to decibels (dB) relative to baseline.

Of the 14 subjects, six were excluded from further EEG analysis during preprocessing. Four subjects were excluded because their EEG data were recorded without the use of a digitization system. Their data could thus not be accurately source localized. Two more subjects were excluded from further EEG analysis due to substantial electromyographic artifact, which resulted in the rejection of the vast majority of trials following the quality assurance procedures described above. The EEG data of the remaining eight subjects was then subjected to source localization and all further analysis described below.

**EEG source localization**. We reconstructed subjects' cortical surfaces from pre-surgical T1 MRI images using Freesurfer v5.3[60]. The EEG cap digitization was manually co-registered to the Freesurfer anatomical reconstruction using the MNE command line tools package. Then, in MNE-Python, the cortical meshes were downsampled from ~160,000 vertices per hemisphere to 4098 dipole locations (vertices) per hemisphere. We calculated a forward solution using the three-compartment boundary-element model[61] with the inner and outer skull surfaces reconstructed from Freesurfer's watershed algorithm[62]. The dipole amplitude (current source density) at each cortical location was estimated using the anatomically constrained MNEs method[63], using a pipeline similar to other reports of region of interest (ROI)-based oscillatory analyses[64]. Briefly, the MNE method finds the maximum a posteriori estimates of the latent cortical sources, given the observed sensor sources, assuming (1) the current source amplitudes are sparse and normally distributed with a known source covariance matrix and (2) the observed sensor data contain additive noise with a normal distribution and a known spatial

covariance matrix. Importantly, as opposed to other beamforming methods, the MNE method preserves oscillations such that oscillatory power can be estimated following source localization. Each vertex's current source estimate includes a dipole orientation, such that the source time course may be either positive or negative at any given time. Here, the orientations of the dipoles were constrained to the cortex using recommended default parameters (loose = 0.2, depth = 0.8). The noise covariance matrices necessary for source localization were estimated per subject from a baseline of period of 500 ms prior to the start of each trial. The empirical covariance estimates were regularized via the "shrunk" method, as recommended by Engemann and Gramfort[65]. Individual source estimate data were then mapped to Freesurfer's "fsaverage" cortical surface. Finally, source estimate time courses for individual vertices were combined within a set of cortical labels corresponding to our ROIs: cingulate cortex (rACC, dACC, mCC), dorso-mPFC (dmPFC/superior frontal gyrus), dorso-lateral prefrontal cortex (DLPFC/middle frontal gyrus), and ventrolateral prefrontal cortex (VLPFC/inferior frontal gyrus). The average time course per ROI was computed using the "PCA flip" technique in MNE-Python. Briefly, singular value decomposition (SVD) is applied to the vertex-wise time courses per ROI and the first right singular vector is extracted. Each vertex's time course is then scaled and sign flipped. The scaling is performed in order to match the average power of vertex-wise time courses. The sign of the time course is adjusted by multiplying it with the sign of the left singular vector from the SVD, which ensures that the phase does not change by 180 degrees from one source time course to the next. Supplementary Table 1 lists these labels and the anatomical shorthand used for each in the main text/figures. The anatomical labels/ROIs were manually assembled by merging of multiple smaller, contiguous labels from the Lausanne 243-region atlas[66]. The labels used here were designed to ensure that each cortical region corresponded to a nearly equal number of vertices in the standard template brain. We selected the label set to cover regions previously implicated in functional neuro-imaging of the MSIT[13,25].

**Statistical analysis—behavior**. The primary behavioral outcome in MSIT is subjects' RT, as they are pre-trained to very low error rates. Along with others, we have shown that RTs during conflict and decision-making tasks are better approximated by gamma than by Gaussian distributions[13,67]. We thus analyzed behavior in a mixed effects GLM with the gamma distribution and identity link function. That GLM was applied at the per-trial level, allowing us to model the effects of DBS and trial-specific effects such as emotion and cognitive interference. The mixed effects design, which includes a random intercept for the subject, specifically controls for intra-subject correlation (trials and sessions as repeated measures). We excluded trials with missing responses, error trials, and post-error trials. We further excluded trials with outlier RTs, which we defined by fitting a gamma distribution to each subject's RT data, pooling the DBS ON and OFF runs for this preprocessing step. We then excluded trials with RT likelihood <0.005 based on the fitted distribution. These approaches excluded 247 trials (6.12% of total, $n = 3785$ trials retained in analysis).

To control for overall RT variability across subjects, we specified GLMs with a subject-specific random intercept plus fixed effects for experiment variables (mixed models). Similar to prior reports, e.g.[28], we identified the appropriate model by minimizing Akaike's information criterion (AIC) during stepwise addition of variables. Importantly, AIC minimization is mathematically equivalent to constructing the model by out-of-sample cross-validation[36], an approach we have identified as essential in biomarker research[38]. We considered interference, DBS, valence, and arousal as possible RT predictors based on our pre-specified hypotheses and the task design. We also tested interaction terms between these main effects. We considered trial number within a run as a nuisance regressor, controlling for fatigue and/or learning effects. The data were best explained by a model with the aforegoing main effects, but no interaction terms (see main text and Supplementary Fig. 1). Models with other predictors, e.g., RT on the preceding trial (an autoregressive effect), were not identifiable. Conflict and DBS were dummy coded, whereas valence, arousal, and trial number were treated as continuous variables. All independent variables were standardized to the 0–1 interval for regression, but are reported in the article after conversion back to their natural units for ease of interpretation.

**Statistical analysis—EEG modulation by task variables and DBS**. For the time-domain (evoked potential) analysis, sensor and source space time courses were reduced to the (−0.5, 2.0) s time window for stimulus-locked epochs and (−1.0, 1.0) windows for response-locked epochs. Furthermore, all epochs were low-pass filtered to 15 Hz and downsampled by a factor of 3. Confidence intervals on plotted event-related potentials (ERPs) were calculated by 1000 bootstrap resamplings with replacement (preserving the number of trials within each subject). All ERPs shown are the grand mean across all subjects.

For the spectral-domain analysis, we calculated non-phase-locked power in three bands of interest: theta (4–8 Hz), alpha (8–15 Hz), and beta (15–30 Hz). We emphasized non-phase-locked, or induced, oscillations because they appear to be more directly related to proactive cognitive control[17]. In trial-based analyses of Simon-effect tasks, over 80% of the conflict/control-related theta power change was non-phase-locked[23]. The non-phase-locked theta power was correlated with trial-to-trial RTs, more so than the phase-locked theta reflected in the time-domain ERP. Further, in a non-trial-structured cognitive control task, theta oscillations

appeared to be continuously present over mid-frontal cortex, increasing in power when more control was needed[68]. In contrast, phase-locked theta oscillations may be more related to error-related performance monitoring[69], a phenomenon not studied here due to the very small number of error trials.

To calculate non-phase-locked power changes, we first subtracted the mean ERP from each trial[23]. The subtracted ERP (and the trials from which it was subtracted) were calculated for each combination of subject, condition (DBS ON/ OFF × Interference/Conflict trials), and ROI/sensor. All plots of EEG power show data after this ERP removal.

Sensor and source-localized data were then decomposed into their time-frequency representation via Morlet wavelet convolution. Wavelets had base frequencies sampled from 2 to 50 Hz in 25 logarithmically spaced steps, where each wavelet was characterized by three cycles. Decomposition was performed on single-trial data, not on the average or ERP. All frequency power estimates were normalized to the average power of a pre-stimulus baseline ($-0.5$ s to $-0.1$ s) for each frequency band. We used a dB transform for normalization. The baseline power was computed separately for each subject and DBS condition (OFF, ON). The same pre-stimulus baseline period used for stimulus-locked analyses was also used for response-locked analyses. We then averaged the values within each pre-specified frequency band to obtain a per-trial power time course for each band. All resulting power values shown in the article were normalized to dB as noted above. All power topographic and time course plots represent the grand mean across subjects.

In both sensor and source space, both time-domain and frequency-domain EEG data were analyzed using ordinary least squares regression[70,71]. The single-trial voltage or power at each time point was entered into a linear model using the same independent variables as the behavioral GLM: interference, DBS, valence, arousal, and trial number. We standardized all independent variables to the [0, 1] interval for this model also. We also considered the possibility that interference and DBS might interact at the neural level even though we saw no behavioral interaction, and thus included a DBS × interference interaction term in this regression. To replicate the effect of the subject-specific intercepts in the behavior model, we subtracted each individual subject's all-trials mean voltage or power time course from that subject's trials. Contrast statistics ($t$-tests) were computed for each resulting beta weight (regression coefficient) at each sample. To control for multiple statistical comparisons (timepoints) within each ROI/electrode, we performed permutation inference and temporal cluster correction[72]. We used 1000 permutations for each analysis, discarded clusters <50 ms in temporal extent, and retained only clusters that were significant at $\alpha = 0.05$. For the time-domain analysis in source space, we further corrected these cluster $p$-values using the Benjamini–Hochberg false discovery rate (FDR) step-down procedure across all tested ROIs. For frequency-domain analysis, we did the same, but using a single step-down across ROIs and frequency bands simultaneously. All significant clusters shown in the article survived these corrections. The exception is that for sensor-space analysis, we did not correct for multiple sensors, because we tested only one sensor for time-domain and one sensor for frequency-domain analysis. The sensor-space frequency-domain $p$-values were again corrected for multiple bands.

**Statistical analysis—EEG/behavioral changes as biomarkers**. We hypothesized that both theta band EEG and MSIT behavior changes induced by DBS might correlate with subjects' clinical response to VCVS DBS treatment. We further hypothesized that this correlation might be with positive clinical response (improvement in depression) or with clinical complications (hypomania, as in[28]). We quantified these at the individual subject level: MSIT RT as the mean (DBS ON–DBS OFF) difference, and theta EEG as the integrated height of the (DBS ON–DBS OFF) difference wave in the VLPFC (anterior inferior frontal gyrus). The VLPFC label was selected as the predictor variable after viewing the results of the preceding analyses. The difference wave was specifically calculated over the time period where we found a significant cluster during the source space analysis. Depression was measured with the Montgomery–Åsberg Depression Rating Scale (MADRS) as collected during the subjects' original clinical trials; we did not attempt correlation with OCD symptoms because only two subjects in the sample had OCD. We used the MADRS change from the pre-implant baseline to the day of data collection, or to the nearest clinical visit to the data collection (always within 1 month) if a given subject was unable to complete the MADRS that day. Hypomania used the same dataset as[28], in which the presence/absence of hypomanic episodes had been coded for each subject by trained clinical raters. The dependent variable was whether that subject had ever had hypomania during their DBS treatment course. One subject was not included in hypomania analyses due to unavailability of clinical data.

Out-of-sample prediction capability is important to assess for putative psychiatric biomarkers[37,38], but difficult to measure in rare populations such as DBS patients. As a surrogate, we generated confidence intervals for the clinical/ biomarker correlations by drawing 1000 bootstrap resamples (with replacement) from the original subject population. We used those same bootstrap draws to construct the confidence interval of the area under the curve (AUC) for receiver-operator characteristic (ROC) curves for classification of hypomania present/absent and depression responder/nonresponder. The latter used the same threshold of 50% MADRS improvement as in the clinical trials, e.g. in[49].

**Statistical analysis—resting-state data**. Theta changes observed during MSIT performance might not be specific to the task, but might arise from a general shift in the EEG frequency spectrum during DBS. Five subjects contributed at least 2 min of eyes-open resting-state data with DBS ON and OFF. From these data, we cut 60 1-s artifact-free epochs from the ON and OFF recordings in each subject, then computed a power spectral density (PSD) from 0 to 30 Hz via the multitaper method. We computed mean power within the theta (4–8 Hz) region of each epoch's PSD, then tested the difference between these distributions with the Mann–Whitney $U$-test. We carried out these analyses on theta power from sensor Fz, which was the scalp point of highest theta power during MSIT performance.

**Validation of MSIT behavioral results in epilepsy controls**. A potential concern is that any RT results we observe might be explainable by practice effects. Although the ON and OFF blocks were separated by an hour or more, subjects might still retain some procedural memory of the task. To address this confound, we analyzed data from a group of subjects who performed multiple temporally spaced MSIT runs without the emotional distractors. These subjects were part of a larger study focused on the network-level physiology of mental illness[13]. They were admitted for inpatient electrophysiologic monitoring of medication-refractory epilepsy. While inpatient, they were approached daily to perform multiple cognitive tasks, including MSIT. In this case, we used the original version of the task, which does not include the background IAPS distractors. Due to the nature of clinical work on an inpatient unit, including breaks for meals and clinical rounds, these subjects often performed one or more 64-trial MSIT blocks with a substantial break in between. This effectively replicates the design of our primary study, except for the DBS manipulation. We analyzed task blocks performed before and after these breaks, in eight subjects. For these subjects, we fit their MSIT trial RTs with a gamma distribution GLM that mimicked the main cohort analysis, i.e., independent/predictor terms for block (which mimics the DBS term), conflict, trial number, and a subject-specific intercept. As with the main cohort, all of these subjects provided full informed consent before any study procedures. All experimental procedures with these subjects complied with governmental and institutional ethics requirements and were approved by the Massachusetts General Hospital Institutional Review Board.

## Data availability

Pre-processed but not analyzed EEG data (source time courses and sensor-space data) and related MRI files is deposited at https://openneuro.org/datasets/ds001784.

## Code availability

Analysis scripts are similarly archived at https://github.com/mghneurotherapeutics/ EMOTE-afMSIT.

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

## Acknowledgements

We gratefully acknowledge technical assistance from Amanda Arulpragasam, Andrew Corse, Nina Levar, and Tommi Raij with subject recruitment and data collection. Graphical elements in Fig. 1a include photography by Elias Levy re-used under CC-BY. This work was supported by grants from the Brain & Behavior Research Foundation (A.S.W.), Picower Family Foundation (A.S.W., E.K.M.), MIT Picower Institute Innovation Fund (E.K.M.), Defense Advanced Research Projects Agency (DARPA) under Cooperative Agreement Number W911NF-14-2-0045 issued by the Army Research Organization (ARO) contracting office in support of DARPA's SUBNETS Program (A.S.W., D.D.D., E.N.E., S.Z., T.D.), Office of Naval Research MURI N00014-16-1-2832 (E.K.M.), and National Institutes of Health (R21MH109722, A.S.W.; R03MH111320, A.S.W. and D.D.D.; UH3NS100548, A.S.W., D.D.D., E.N.E., and T.D.; R37MH087027, E.K.M.). The views, opinions, and findings expressed are those of the authors. They should not be interpreted as representing the official views or policies of the Department of Defense, Department of Health & Human Services, any other branch of the U.S. Government, or any other funding entity.

## Author contributions

A.S.W., D.D.D and E.K.M. designed the research. T.D. designed the behavioral task. E.N.E. performed neurosurgical procedures in the subjects' clinical trials. A.S.W., D.D.D and E.N.E. obtained research funding and provided general team supervision. A.S.W., A.C.P., I.B., and S.S.C. collected data in epilepsy subjects. A.S.W. and S.Z. collected EEG/behavioral data, performed the analyses, and created the figures. A.S.W., S.Z. and E.K.M. wrote the manuscript. All authors had an opportunity to review the manuscript and revise for critical intellectual content.

## Additional information

**Competing interests:** A.S.W., D.D.D., E.K.M., E.N.E., and T.D. are named inventors on patent applications related to deep brain stimulation and oscillations, including at least one application related to the subject of this paper. A.S.W., D.D.D., and E.N.E. have received consulting income and/or research support from Medtronic, which manufactured the devices used in this study. Medtronic had no financial or technical involvement with this specific research. T.D. discloses honoraria, consultation fees and/or royalties from the MGH Psychiatry Academy, BrainCells Inc., Clintara, LLC., Systems Research and Applications Corporation, Boston University, the Catalan Agency for Health Technology Assessment and Research, the National Association of Social Workers Massachusetts, the Massachusetts Medical Society, Tufts University, NIDA, NIMH, and Oxford University Press. None of those entities manufactures technology or products used in the study. The remaining authors declare no competing interests.

