## [Peer Review File · Nature Communications]

Reviewers' comments:

Reviewer #1 (Remarks to the Author):

This is a well-written manuscript reports on findings of effects of ventral internal capsule/ventral striatum deep brain stimulation on cognitive control. The finding of increased frontal theta power and enhanced cognitive control after DBS is important and novel and certainly deserves publication. The methods are sufficiently explained and sound. The interpretations however, are not supported by the data presented.

- Page 3: "Without knowledge of DBS' mechanism of action, study clinicians do not know whether any individual patient's stimulating field is properly adjusted to activate the necessary circuits and achieve the desired results". This is a commonplace statement. only very few medical treatment benefit from knowing exact effects bur still work.

- It has been demonstrated in 2007 that DBS to the STN selectively interferes with the normal ability to slow down when faced with decision conflict (PMID: 17962524). Could the improved cognitive control reported in the manuscript be explained by an increase of impulsivity? Were there any attempts to control for impulsivity?

- A paper in 2012 (PMID: 22414813) posited theta power as a predictor of response in SCC DBS; why is it not discussed? SCC is an altogether different target, this would speak against the speciality of VCVS as discussed in the manuscript.

- Patients suffering from major depression and OCD are included. Effects of disease and eventual response status on theta power are likely, but not controlled for or even discussed.

- Where there disease specific differences in the measures obtained?

- Cognitive control dysfunctions are thought to contribute to the onset and maintenance of depression, OCD, substance abuse and probably many other neuropsychiatric disorders. It is a huge jump to conclude from results of cognitive control enhancement in 2 OCD and 12 MDD patients that DBS to VCVS is the key to the "mechanistic understanding" of these disorders.

Reviewer #2 (Remarks to the Author):

This paper reports an experiment in which electroencephalography (EEG) was measured during a cognitive control task in a group VCVS DBS-implanted patients with major depression (n = 12) or OCD (n=2), after 0.5-7 years of chronic stimulation, and after 1 hour of stimulation discontinuation. During active DBS, response times were 34ms shorter, not explained by psychomotor -or error-rate changes and therefore suggestive of improved cognitive control. In addition, in a subset of 8 patients DBS increased task-related theta power in the VLPFC, which correlated with DBS-induced depression improvement. Though based upon a small number of patients, which is unavoidable in studies with psychiatric DBS patients, these results are important for the development of DBS for psychiatry by demonstrating a promising electrophysiological and potentially cross-diagnostic antidepressant mechanism of VCVS DBS. The experimental setup and methods are clever and precise, although there are some limitations that may require additional clarification or improvement.

1. A potential limitation is the lack of a control group. Without repeated measures in non-DBS controls, it remains uncertain whether the current findings are truly related to DBS, and if they are indicative of improvement/ normalization.

2. An obvious limitation is the small number of subjects. Although 14 patients performed the tasks, only for 8 patients task-related EEG results were available. It must be noted that these

numbers are common for these kinds of extremely difficult experiments in rare DBS-implanted psychiatric patients, and the authors have acknowledged this limitation in their discussion.

3. Another question is about harmonic artifacts. The authors may want to present the frequency of stimulation for each patient or perhaps even calculate the expected harmonics, which will likely include bands under the low pass filter they selected (50 Hz). Supplemental Figure 4B is an important proof related to this, but it is based on data for only 4 patients. It could be one patient driving the reversal, which is key to the interpretation of the main results. Please consider completing ON/OFF resting-state recordings in the entire sample of 14 patients or mention this limitation.

4. Could the authors clarify the spectral decomposition method. Specifically, how were the source waveforms transformed using Morelet wavelets? Source results are not oscillatory - there is no negative going component. This is a non-trivial methodological challenge with spectral analysis of EEG sources. A combination of spectral decomposition and PCA on the scalp surface could be considered, with source analysis of selected components.

5. The spectral data has the ERP subtracted from it, therefore representing non-phase-locked spectral changes. Please provide a rationale or interpretation of that, as there is much debate on this topic and it would be useful to the readers to have this information.

6. In figures 1C&D, bars that show SD are not error bars, why did the authors not choose presenting SE of the mean? Moreover, instead of main effects, the expected result to present would be the interaction cognitive control/interference X on/off.

7. The authors' statement that based on animal research 1 hour of DBS-discontinuation may be enough to overcome rebound effects should be nuanced, as at least one previous study in OCD-patients suggested rebound effects after 1 week of DBS-discontinuation (Ooms et al., 2014).

8. In the discussion, the authors mention that DBS-induced theta change was associated with a smaller clinical effect, which I do not fully understand given the positive correlation with depression improvement?

9. The authors discuss the potential of future stimulation parameters directly titrated to change a theta biomarker or using theta for self-titrating stimulation. These potential applications may be discussed in more detail given that DBS in this study was able to either reduce (at rest) or enhance (during cognitive control) prefrontal theta.

Reviewer #3 (Remarks to the Author):

The authors report on the effect of VC/VS DBS on cognitive control in 14 patients with MDD/OCD. They report that their results show that VC/VS DBS enhances cognitive control and that this effect is mediated by the ability of VC/VS DBS to enhance PFC theta oscillations. The authors' claim is that this type of neuroanatomic/neurophysiologic "mechanistic" finding "may be a path to understanding therapeutic mechanisms". This is a worthy and rational effort, but the authors present many assumptions in the development of the study that require stronger support and employ some methods that undermine their central claims.

The fundamental weakness in this study is the grouping of data linking the individual patients' cognitive control/PFC theta findings to each patients' therapeutic DBS response. The authors provide only indirect evidence that their results are relevant to the central issue of how VC/VS DBS effectively treats MDD, or how one might optimize an individual's DBS placement or stimulation parameter settings to achieve maximum therapeutic benefit. The repeated measures group data design obscures the very link the authors are purportedly interested in, that is, the link between individual variability in the DBS therapeutic response, and variability in DBS-enhanced cognitive control/PFC theta oscillations

Some general issues that the authors should address in more detail or correct:

The concept of "cognitive control" as a potential psychological factor underlying MDD/OCD is far broader than the very narrow operationalization employed here as "greater cognitive control" = "faster response times on the MIST". The authors should provide greater justification for this

operationalization and its relevance to the broad concept of "cognitive control".

It is misleading to state that previous DBS trials that do not demonstrate or confirm the effectiveness of a particular DBS approach for a particular condition/indication have "failed". The purpose of DBS trials is to determine whether a particular DBS approach in a specific target location for a specific indication may be effective or useful, not to show that the researchers' assumptions are confirmed.

It is too strong or simply incorrect to state that previous trials of DBS were conducted without knowledge of the mechanisms of DBS. The electrical stimulation of neural tissue has been studied for decades. In 1975, Ranck provided a useful summary of the mechanisms of electrical stimulation on the mammalian central nervous system. In the current era of human DBS, there are hundreds of studies on its mechanisms. While there is much debate on this topic and more work is needed to fully elucidate the specific mechanisms in specific applications, the oft-repeated misconception that "no one understands how DBS works" is inaccurate and should not be perpetuated here.

The PFC is an imprecise term. The authors should attempt to specify the specific brain region and associated specific brain circuit throughout the paper wherever possible.

Specific issues:

In Fig 1D, there appears to be no significant difference in response time with DBS on or off. The 1 SD error bars completely overlap. Yet, in the text, the authors state that response times were significantly decreased with DBS-on. Similarly, in Fig 2 E and F, the error bars overlap for the mCC, yet the authors report significant differences. This all needs to be clarified.

Even if response times were faster with DBS on and DBS increases PFC theta, it is a fundamental error to state, as the authors do, that this correlation "suggests that PFC theta oscillations play a causal role in effective cognitive control, and that amplifying those oscillations facilitates control." If on-off RT changes did not correlate with patients' clinical DBS response (Fig 3A-B), and RT is the operational surrogate for "cognitive control", don't these results undermine the authors' claim for the central importance of cognitive control as a DBS "target" in treating MDD/OCD?

Why were 13 patients included in the analysis of on-off RT changes (Fig 3A-B), but only 8 patients included in the analysis of theta changes (Fig 3C-D)? Why not all 14 patients for each analysis?

What are the results when conducted using the data from all 14 patients?

The authors claim that hypomania is a manifestation of DBS-induced impulsivity, and that the lack of correlation between theta or RT and hypomania shows that the results are not due to DBS-induced impulsivity is tenuous and should be either more strongly supported or eliminated.

We thank the reviewers for a set of clear and well-reasoned critiques that we believe have made the paper stronger. We reply to each individual item below and have made substantial changes to the manuscript to address the concerns. These are highlighted through Microsoft Word's "track changes" functionality.

Reviewer #1 (Remarks to the Author):

This is a well-written manuscript reports on findings of effects of ventral internal capsule/ventral striatum deep brain stimulation on cognitive control. The finding of increased frontal theta power and enhanced cognitive control after DBS is important and novel and certainly deserves publication. The methods are sufficiently explained and sound. The interpretations, however, are not supported by the data presented.

We agree that the language in some places was overly strong, and have made adjustments in line with your comments. See below for specifics.

- Page 3: "Without knowledge of DBS' mechanism of action, study clinicians do not know whether any individual patient's stimulating field is properly adjusted to activate the necessary circuits and achieve the desired results". This is a commonplace statement. only very few medical treatment benefit from knowing exact effects bur still work.

Absolutely true. We believe, however, that this is a particularly pernicious problem for brain stimulation in psychiatry. For other treatments, such as psychotropic medication, one can at least assess receptor occupancy or serum levels. For DBS in applications such as Parkinson disease, tremor is a visible and objective sign whose suppression can be verified. The programming of psychiatric DBS is far more subjective and prone to error. We describe this in the cited references and in (Widge AS, Dougherty DD. Managing patients with psychiatric disorders with deep brain stimulation. In: Marks Jr. WJ, editor. Deep Brain Stimulation Management. 2nd ed. Cambridge : New York: Cambridge University Press; 2015.) For this reason, we believe that there is a unique need here for mechanistic understanding.

Your point remains that we did not articulate this very well, and we have edited the Introduction (pages 3-4) to elaborate better.

- It has been demonstrated in 2007 that DBS to the STN selectively interferes with the normal ability to slow down when faced with decision conflict (PMID: 17962524). Could the improved cognitive control reported in the manuscript be explained by an increase of impulsivity? Were there any attempts to control for impulsivity?

Thank you for the reminder; we were aware of the work of Frank et al. and this specific paper, but committed an oversight in not citing it. It has been added.

To your larger question, we do not think the results are explained by impulsivity, and we specifically point to the results of Supplementary Figure 2 as a control. We ran the Effort Expenditure for Rewards

Task (EEfRT) directly after the Multi-Source Interference Task, in both DBS ON and DBS OFF conditions. In EEfRT, subjects are asked to choose between an easy button-pushing task (with fixed low payoff) or a harder task (with variable high payoff). While not identical, this has some conceptual overlap with the “Win-Win” conflict situations in the Frank et al. paper and the 2011 follow-on by Cavanagh et al. In those papers, DBS ON (at STN) caused faster choice reaction times specifically during conflict. We did not observe that effect -- instead, we observed subjects to be significantly slower during EEfRT with DBS ON. We have added a new panel C to Supplementary Figure 2 to demonstrate that this is true at the group and single-patient level. We also note that, in contrast to the Frank/Cavanagh results, the effect of VCVS DBS was not limited to conflict trials (see new panel in Supplementary Figure 1).

We have expanded the Discussion (p14) to summarize these points.

- A paper in 2012 (PMID: 22414813) posited theta power as a predictor of response in SCC DBS; why is it not discussed? SCC is an altogether different target, this would speak against the speciality of VCVS as discussed in the manuscript.

You are right that this is worth discussing. We specifically mentioned this paper from Broadway et al. in a recent meta-analysis (Widge et al., *American Journal of Psychiatry* in press, doi:10.1176/appi.ajp.2018.17121358) that also discussed theta biomarkers. We do note that the results of the Broadway et al. paper do not quite align with what we are reporting. That study used resting state theta (which we found to decrease or remain unchanged, see further below), not the task-evoked theta we describe here. It also applied a derived theta cordance measure, not actual theta power. These are very different phenomena. Most importantly, the primary analysis in that paper (ANOVA of theta cordance with independent variables of responder/nonresponder status and assessment timepoint) did not meet the pre-specified significance threshold. Even allowing for that limitation, the authors report that cordance specifically increased in the responder group with SCC DBS treatment. In all other studies that have claimed frontal theta cordance as a biomarker, the signature of response is a decrease with treatment.

That said, your question highlights that this is another topic we could/should have included in the paper and discussed in more detail, and we have now done so on p14-15, with reference to the original Broadway paper, a related review by Pizzagalli, and our meta-analysis.

- Patients suffering from major depression and OCD are included. Effects of disease and eventual response status on theta power are likely, but not controlled for or even discussed.
- Where there disease specific differences in the measures obtained?

There was no evidence that these measures, and in particular the relation between neurophysiologic change and clinical outcome, differed by primary diagnosis. In a small and unbalanced sample, we are not powered to perform any type of statistical comparison, but have marked the two subject types differently in Figure 3. The OCD and MDD subjects can be seen to be intermixed in all panels of this figure. This was a very reasonable question and we have updated the textual discussion of Figure 3's results to highlight the point.

- Cognitive control dysfunctions are thought to contribute to the onset and maintenance of depression, OCD, substance abuse and probably many other neuropsychiatric disorders. It is a huge jump to conclude from results of cognitive control enhancement in 2 OCD and 12 MDD patients that DBS to VCVS is the key to the "mechanistic understanding" of these disorders.

We agree. We have, throughout the Introduction and Discussion, attempted to soften language to emphasize that this is one of several possible mechanisms (and that VCVS DBS itself may have pleiotropic effects). As far as we are aware, the paper as resubmitted does not contain the words “the key” or similar statements of definitive proof.

Reviewer #2 (Remarks to the Author):

This paper reports an experiment in which electroencephalography (EEG) was measured during a cognitive control task in a group VCVS DBS-implanted patients with major depression (n = 12) or OCD (n=2), after 0.5-7 years of chronic stimulation, and after 1 hour of stimulation discontinuation. During active DBS, response times were 34ms shorter, not explained by psychomotor -or error-rate changes and therefore suggestive of improved cognitive control. In addition, in a subset of 8 patients DBS increased task-related theta power in the VLPFC, which correlated with DBS-induced depression improvement. Though based upon a small number of patients, which is unavoidable in studies with psychiatric DBS patients, these results are important for the development of DBS for psychiatry by demonstrating a promising electrophysiological and potentially cross-diagnostic antidepressant mechanism of VCVS DBS. The experimental setup and methods are clever and precise, although there are some limitations that may require additional clarification or improvement.

1. A potential limitation is the lack of a control group. Without repeated measures in non-DBS controls, it remains uncertain whether the current findings are truly related to DBS, and if they are indicative of improvement/ normalization.

We agree. We have addressed this by adding further control data from a non-psychiatric population. We have been using the same Multi Source Interference Task (without the emotional distractors, which in the present study showed little effect) in a cohort of subjects undergoing intracranial monitoring before surgery for intractable epilepsy. Like the subjects in the current study, the subjects in the epilepsy study performed multiple MSIT blocks with long breaks in between blocks. Unlike the subjects in the present study, the subjects in the epilepsy study, who received no brain stimulation, had no change in their response times between blocks. These findings are discussed in the text and included as Supplementary Figure 3. We believe this strongly implicates DBS as the causative agent in our observed effects.

In addition, we note an important point that further addresses your concern. The most likely change, in the absence of perturbation such as brain stimulation, would be a practice effect. This would manifest as OFF

< ON, because we did the OFF block second. We observed the precise opposite. We have emphasized this point in the text as well.

2. An obvious limitation is the small number of subjects. Although 14 patients performed the tasks, only for 8 patients task-related EEG results were available. It must be noted that these numbers are common for these kinds of extremely difficult experiments in rare DBS-implanted psychiatric patients, and the authors have acknowledged this limitation in their discussion.

Thank you for your recognition of the extreme challenges of research with this population and the relatively large sample size here relative to the universe of available subjects. We agree this is a limitation and have firmly acknowledged that point, as well as highlighted the need for prospective replication (p12-13).

3. Another question is about harmonic artifacts. The authors may want to present the frequency of stimulation for each patient or perhaps even calculate the expected harmonics, which will likely include bands under the low pass filter they selected (50 Hz). Supplemental Figure 4B is an important proof related to this, but it is based on data for only 4 patients. It could be one patient driving the reversal, which is key to the interpretation of the main results. Please consider completing ON/OFF resting-state recordings in the entire sample of 14 patients or mention this limitation.

A fair point. Regarding harmonics, these are not a concern because of the stimulation frequency. With the exception of one subject, stimulation was at or above 100 Hz. The patient who was an exception had stimulation at 50 Hz. We specifically selected our lowpass filter to ensure that the stimulation artifact was not present in any of the analyzed EEG data. We have clarified this point in the text and have added the subjects' stimulation frequencies to Supplementary Table 1. We note also that we have successfully used this same approach, as have others, to analyze DBS-contaminated EEG (Widge, Zorowitz, et al., *Biological Psychiatry* 2016, plus companion article by Bahramisharif et al., same issue). That paper also shows resting state spectra demonstrating no narrow-band peaks that would suggest DBS harmonics.

Regarding the question of whether Figure S4 (now Figure S5)'s results are influenced by a single subject, we have now included the individual subject resting spectra in that figure to address your concern. It appears that this concern was well founded; only two subjects showed a clear theta-band increase with DBS OFF. The others who contributed data showed no change. In the process of performing this analysis, we also identified one subject who did not contribute usable task-related EEG, but did contribute usable resting state data. We have added that subject to the analysis. With this change, the resting-state spectra are now identical in the ON and OFF condition, and we have revised the manuscript appropriately.

We agree with your suggestion that the ideal experiment would be to re-collect data on all 14 subjects. We are unable to do so because (A) the primary experimenters and data analysts are no longer at the same institution as the subjects and (B) about half of the subjects have since undergone follow-up neurosurgeries and/or DBS explanations, making them unavailable for data contribution. As you noted in

your prior point, this is generally a limitation of the study that we have done our best to acknowledge. Regardless, we no longer make a claim about change in resting state spectrum.

4. Could the authors clarify the spectral decomposition method. Specifically, how were the source waveforms transformed using Morelet wavelets? Source results are not oscillatory - there is no negative going component. This is a non-trivial methodological challenge with spectral analysis of EEG sources. A combination of spectral decomposition and PCA on the scalp surface could be considered, with source analysis of selected components.

We agree this is a point worth clarifying. Source results can have oscillatory activity, i.e. both positive and negative values, depending on the transform that is applied to the sensors. There are two specific aspects of the inverse method we used that preserve oscillations. First, MNE and its dSPM method permit the dipole orientation at each cortical vertex to vary, such that the dipole can be pointed into or out of the brain. This, combined with assignment of a sign to the current source density at each timepoint, preserves oscillations. Second, we combined vertices within cortical labels using MNE's pca-flip method. This effectively combines vertices by weighting them with the first eigenvector of the SVD of the vertex x time matrix. In doing so, it can assign positive or negative weights, s.t. the combination of individual vertices at any given timepoint can be positive or negative. In this way, oscillatory activity at the scalp is preserved. To further verify this, both for your review and for readers, we have provided plots of this oscillatory activity in Supplementary Figure 3 and have expanded the Methods.

5. The spectral data has the ERP subtracted from it, therefore representing non-phase-locked spectral changes. Please provide a rationale or interpretation of that, as there is much debate on this topic and it would be useful to the readers to have this information.

This could indeed have been discussed better. Our analysis here was heavily influenced by the work of Cohen and colleagues (particularly, Cohen & Donner 2013), who have presented an argument that the type of stimulus-locked, conflict-related theta we are studying specifically represents a change in oscillatory power without a trial-to-trial phase resetting. This has been backed up by other studies, as reviewed both by Cohen and by Cavanagh & Frank (2014). We have added a paragraph to the Methods discussing this past work and the rationale for our analytic approach. In Supplementary Figure 3, we also show evidence that the majority of task-evoked theta was in fact non-phase-locked.

6. In figures 1C&D, bars that show SD are not error bars, why did the authors not choose presenting SE of the mean? Moreover, instead of main effects, the expected result to present would be the interaction cognitive control/interference X on/off.

Two reviewers raised this point. We have revised plots throughout the paper to show SEM, except in instances where we already displayed an SEM equivalent (i.e., the bootstrapped confidence interval of Supplementary Figure 4). We have also added the requested plot of interaction effect, as a panel in Supplementary Figure 1. We respectfully disagree, however, that the interaction is the outcome of interest. In the task structure we chose, where conflict/interference trials are frequent and the trial

structure is unpredictable, the neural circuitry of cognitive control must always be “on alert”. That is predicted in part by Shenhav (2016)’s formulation of control itself as effortful/costful in a way that brain homeostatic processes then attempt to minimize. We see this in, e.g. Supplementary Figure 5 (panel A), where theta increases over baseline even during non-interference trials. In that context, if DBS is augmenting the function of that neural circuitry, we would expect theta increases and reaction time decreases in all trials, not exclusively in interference trials. That is precisely what we observed. You are right that we need to clarify why we expected this result and what it means, and we have expanded the Discussion (p15) accordingly.

7. The authors’ statement that based on animal research 1 hour of DBS-discontinuation may be enough to overcome rebound effects should be nuanced, as at least one previous study in OCD-patients suggested rebound effects after 1 week of DBS-discontinuation (Ooms et al., 2014).

Quite right. We do not think the rebound is over at 1 hour -- we believe it is just beginning, and that the rebound effects were persistent throughout our entire DBS OFF period. The study is designed to capture this rebound and to use it to emphasize how DBS is changing the brain. Put another way, we believe that the phenomena we observe are an exaggeration of the actual change induced by chronic DBS, but an exaggeration that helps to highlight the putative physiologic mechanism. We have clarified this point (and cited the Ooms paper) in the Methods.

8. In the discussion, the authors mention that DBS-induced theta change was associated with a smaller clinical effect, which I do not fully understand given the positive correlation with depression improvement?

In Figure 3, a larger ON-OFF theta change is associated with a smaller improvement in depression (change in MADRS score). The MADRS axis is scored such that negative values indicate better therapeutic response. We have clarified this point in the figure legend. As noted in the Discussion, our suspicion is that this represents an index of neuroplasticity -- that the more quickly theta falls off with DBS OFF, the less the DBS had succeeded in engaging and changing circuits involved in cognitive control. Put another way, it is a measure of how much “vulnerability” remains in the patient’s cognitive control network even with chronic DBS therapy. This is covered in the last paragraph of the Discussion.

9. The authors discuss the potential of future stimulation parameters directly titrated to change a theta biomarker or using theta for self-titrating stimulation. These potential applications may be discussed in more detail given that DBS in this study was able to either reduce (at rest) or enhance (during cognitive control) prefrontal theta.

We have expanded the discussion of this point (p12) and provided reference to some of our other work in support of the general concept.

Reviewer #3 (Remarks to the Author):

The authors report on the effect of VC/VS DBS on cognitive control in 14 patients with MDD/OCD. They report that their results show that VC/VS DBS enhances cognitive control and that this effect is mediated by the ability of VC/VS DBS to enhance PFC theta oscillations. The authors' claim is that this type of neuroanatomic/neurophysiologic "mechanistic" finding "may be a path to understanding therapeutic mechanisms". This is a worthy and rational effort, but the authors present many assumptions in the development of the study that require stronger support and employ some methods that undermine their central claims.

The fundamental weakness in this study is the grouping of data linking the individual patients' cognitive control/PFC theta findings to each patients' therapeutic DBS response. The authors provide only indirect evidence that their results are relevant to the central issue of how VC/VS DBS effectively treats MDD, or how one might optimize an individual's DBS placement or stimulation parameter settings to achieve maximum therapeutic benefit. The repeated measures group data design obscures the very link the authors are purportedly interested in, that is, the link between individual variability in the DBS therapeutic response, and variability in DBS-enhanced cognitive control/PFC theta oscillations

We agree that it is critical to account for the repeated measures design and to show both group-level and individual effects. To the former, we specifically chose a mixed-effects model to account for this. We have emphasized that point in the revised Methods. We have also added further data (see response to Reviewer 2) that demonstrate that our two-run design does not *per se* cause any behavioral effects. To the latter point, the revised Figure 3 specifically shows each individual patient's behavioral and electrophysiologic change in response to the DBS manipulation, plotted against that same patient's clinical response. Supplementary Figure 9 also shows individual-level relationships between theta oscillations and response times.

We further agree that an interesting and necessary follow-up would be to assess theta change and task performance at multiple timepoints for each patient in conjunction with his/her DBS response. We are currently performing that prospective study as part of a larger DBS biomarkers protocol; it is registered on clinicaltrials.gov as NCT03184454. Recruitment should be complete in 2023. We have emphasized the need for further replication, particularly prospectively, in the Discussion.

Some general issues that the authors should address in more detail or correct:

The concept of "cognitive control" as a potential psychological factor underlying MDD/OCD is far broader than the very narrow operationalization employed here as "greater cognitive control" = "faster response times on the MIST". The authors should provide greater justification for this operationalization and its relevance to the broad concept of "cognitive control".

A very fair point, and in the Methods, we now discuss some alternate tasks/formulations that can also be considered as part of the cognitive control construct. In the Discussion (p13), as part of our consideration of next steps, we discuss the need for future studies to assess cognitive control using a broader task battery. Most importantly, in the Methods, we further justify our specific use of MSIT performance as our operationalization in this study. MSIT has been specifically designed and verified to induce strong

subject-level effects, which we felt would be useful for this study. In fact, we believe our use of MSIT helped us elucidate the individual-level results described in our response to your prior point.

It is misleading to state that previous DBS trials that do not demonstrate or confirm the effectiveness of a particular DBS approach for a particular condition/indication have “failed”. The purpose of DBS trials is to determine whether a particular DBS approach in a specific target location for a specific indication may be effective or useful, not to show that the researchers’ assumptions are confirmed.

Absolutely correct. We have made the same “there are no failed trials, we always learn something” point in our recent clinical reviews (e.g., Widge, Malone, and Dougherty, *Frontiers* 2018). We have revised the text to remove references to trial “failures” and have referenced the aforementioned review where we delve into this question in more detail.

It is too strong or simply incorrect to state that previous trials of DBS were conducted without knowledge of the mechanisms of DBS. The electrical stimulation of neural tissue has been studied for decades. In 1975, Ranck provided a useful summary of the mechanisms of electrical stimulation on the mammalian central nervous system. In the current era of human DBS, there are hundreds of studies on its mechanisms. While there is much debate on this topic and more work is needed to fully elucidate the specific mechanisms in specific applications, the oft-repeated misconception that “no one understands how DBS works” is inaccurate and should not be perpetuated here.

You are right that this discussion could have been more nuanced. As you note, we do have some insights and theories, although those theories remain a matter of vigorous debate even in “solved” areas such as Parkinson disease. That said, Ranck’s summary of the effects of single pulses and short trains, as applied to well-isolated neurons where the electrode position is known relative to the soma/axon/dendrites, shed minimal light on that debate. We know that the effects of chronic high-frequency stimulation, applied to large bundles of mixed axons and somata, are very different even from the effects of brief high-frequency stimulation. (For instance, the long timescales required for clinical response in both mental disorders and dystonia, despite the frequent reports of acute intraoperative mood effects from test stimulations.) Mechanistically, recent trials such as RECLAIM and BROADEN involved a hope that DBS would simulate a lesion or inactivation of the targeted brain region. Recent results, whether in the present study or Helen Mayberg’s large body of work or the physiologically-informed stimulation efforts of Tass and colleagues, clearly demonstrate that “virtual lesion” theory to be incomplete, at best.

A full treatment of this evidence is a review in and of itself. We have softened and nuanced the relevant section of the Introduction (p3-4) and have included reference to our recent in-depth review of the mechanistic debate.

The PFC is an imprecise term. The authors should attempt to specify the specific brain region and associated specific brain circuit throughout the paper wherever possible.

We have done so. This is challenging because in many cases, we saw DBS-induced changes across multiple frontal gyri that could reasonably be labeled “PFC”. All of these structures project through the ventral internal capsule, as documented by Susanne Haber and colleagues. We have endeavoured to be clear about cases where effects were more anatomically restricted, and where we retained the general term “PFC” we have qualified it by being clear that the effects were broad.

Specific issues:

In Fig 1D, there appears to be no significant difference in response time with DBS on or off. The 1 SD error bars completely overlap. Yet, in the text, the authors state that response times were significantly decreased with DBS-on. Similarly, in Fig 2 E and F, the error bars overlap for the mCC, yet the authors report significant differences. This all needs to be clarified.

The error bars did overlap, because we plotted 1 SD, which is not the same as the 95% confidence intervals of the corresponding regression coefficient/group means. We have replaced these with SEMs as noted above, in response to another Reviewer. As seen in Figure 1 and elsewhere, error bars do not overlap.

For Figure 2, there is an important point -- the stars denoting significance are not directly derived from the bars in panels E and F. Rather, they are derived from the more rigorous FDR-corrected significance clusters shown in panels C and D (as grey background) or panel G (as red bars). The bars and SEMs in E-F, on the other hand, are all derived from a fixed time window (the same time window as panels A-B) in order to facilitate comparison across many regions. Thus, significant ON-OFF differences in E-F do not directly track the means and error bars. We have clarified this in the figure legend.

Even if response times were faster with DBS on and DBS increases PFC theta, it is a fundamental error to state, as the authors do, that this correlation “suggests that PFC theta oscillations play a causal role in effective cognitive control, and that amplifying those oscillations facilitates control.”

We have adjusted the sentence to a gentler and less definitive claim, since you are right that there are links in the causal chain that remain to be conclusively proven. That said, we do show theta increases from VCVS DBS, and the correlation between theta and performance on cognitive control tasks (of many kinds, not just MSIT) is a well-established finding. Further, other studies are finding very similar results, e.g. deficits in theta leading to deficits in cognitive control in schizophrenia (Ryman et al., *Biological Psychiatry* 2018). It is difficult to dispute a role for theta in cognitive control given this very large body of prior work, and the DBS-induced theta increase in our study is neither small nor dependent on individual outliers.

If on-off RT changes did not correlate with patients’ clinical DBS response (Fig 3A-B), and RT is the operational surrogate for “cognitive control”, don’t these results undermine the authors’ claim for the central importance of cognitive control as a DBS “target” in treating MDD/OCD?

The results certainly argue that the effect at the neurophysiologic level is more important than the task response time. The cautionary point here is that the clinical response is measured with the MADRS, which is generally recognized to be a blunt instrument that does not entirely align with objective clinical measures. (We discuss this problem further in Widge/Malone/Dougherty 2018, Widge et al. *Experimental Neurology* 2017, and Widge et al. *American Journal of Psychiatry* in-press, and we have ensured that all three are cited as part of this manuscript.) Put another way -- as you have highlighted above, there are many operationalizations of cognitive control. All appear to involve theta power increases, especially in midline PFC regions such as SMA, MCC, and dACC. It is entirely possible that behavior on a different control measure would have correlated with clinical improvement, and that this behavior would have also correlated with the DBS-induced theta increase. That, minus the DBS, is what the Ryman study found using the AX-CPT paradigm.

We have expanded the Discussion (p13) to reflect on and clarify these points. More generally, we do not wish to claim that cognitive control is necessarily of central importance or the sole mechanism by which DBS (at VCVS or any other target) acts. Our results suggest that control and its neurophysiologic underpinnings are a mechanism. We have clarified this point as well in our revisions in response to the other Reviewers.

Why were 13 patients included in the analysis of on-off RT changes (Fig 3A-B), but only 8 patients included in the analysis of theta changes (Fig 3C-D)? Why not all 14 patients for each analysis? What are the results when conducted using the data from all 14 patients?

Not all subjects contributed usable data. Some subjects were not able to withhold verbal responses, prominent facial movements, and/or eyeblinks during task trials, and produced data that could not be adequately cleaned of movement artifact. We have clarified this point and related technical limitations in the Methods.

The authors claim that hypomania is a manifestation of DBS-induced impulsivity, and that the lack of correlation between theta or RT and hypomania shows that the results are not due to DBS-induced impulsivity is tenuous and should be either more strongly supported or eliminated.

We agree that this could be better supported, and have augmented the paper to this end. Specifically, we have included more data from the companion EEfRT task that was performed directly after the MSIT runs, presented in Supplementary Figure 2. Past results showing increased impulsivity from STN DBS make specific predictions about choice response time in “win-win” reward-based tasks like EEfRT. We did not observe that impulsivity signature, and in the Discussion (p14) we now extensively compare and contrast these findings. We also mention them in the Results where we discuss hypomania. We do believe it is reasonable to consider hypomania as one aspect of DBS-induced impulsivity; impulsive seeking of reward and sensation is key to both syndromes.

REVIEWERS' COMMENTS:

Reviewer #1 (Remarks to the Author):

The authors addressed my concerns sufficiently.

Thomas E. Schlaepfer, MD

Reviewer #2 (Remarks to the Author):

The authors have addressed my concerns sufficiently.

They have added new control data from non-psychiatric individuals that, contrary to the DBS patients, had no change in their response times between blocks.

Unfortunately, when including individual resting-state data, theta differences between ON and OFF disappeared, which is adjusted and acknowledged as a limitation in the revised manuscript.

I appreciate the revised explanation regarding oscillations in source results.

Issue of harmonics are still not addressed. That the authors reference their own previous work to argue it doesn't matter isn't particularly strong. However, there is no clear evidence that harmonics are an issue.

The limitations of the 1 hour discontinuation related to rebound are adequately acknowledged.

Reviewer #3 (Remarks to the Author):

This is an important and well-conceived study. It provides a valuable new framework for the design and evaluation of DBS for psychiatric diseases. An unfortunate feature of the revised report is that the authors continue to over-emphasize their claim that alterations in "cognitive control" underlie "mental illness" (MDD, OCD, . . .), even when the measure of cognitive control that they used in this study does not clearly support this. I would encourage the authors to restrict their conclusions to the circumscribed findings drawn from their study results. This issue can be further evaluated in future studies. There are some specific changes I would request, however:

P. 1: Remove "cognitive and physiologic" before "mechanisms of action". Opening up the idea that there are "cognitive" mechanisms of action for DBS unnecessarily enters into conceptually flawed/confused territory that is best avoided in a leading science journal.

The authors mention "noninvasive stimulation" at various points throughout the report, including: "Non-invasive stimulation can enhance cortical oscillations if very carefully designed, and might achieve the same clinical benefits without the risks and expense of neurosurgery." There are many psychotherapeutic, pharmacotherapeutic, and other treatment modalities that could be useful to treat psychiatric disease. This study investigated DBS. Extending the findings from this study to speculations about the possible benefits of other treatment modalities is unnecessary and unsupported.

REVIEWERS' COMMENTS:

Reviewer #1 (Remarks to the Author):

The authors addressed my concerns sufficiently.

Thomas E. Schlaepfer, MD

Thank you, Dr. Schlaepfer.

Reviewer #2 (Remarks to the Author):

The authors have addressed my concerns sufficiently.

They have added new control data from non-psychiatric individuals that, contrary to the DBS patients, had no change in their response times between blocks.

Unfortunately, when including individual resting-state data, theta differences between ON and OFF disappeared, which is adjusted and acknowledged as a limitation in the revised manuscript.

I appreciate the revised explanation regarding oscillations in source results.

Thank you.

Issue of harmonics are still not addressed. That the authors reference their own previous work to argue it doesn't matter isn't particularly strong. However, there is no clear evidence that harmonics are an issue.

We apologize for being insufficiently clear in our response. We agree that harmonics would matter. However, we only analyzed frequencies below the stimulation frequency, in which harmonics would not appear. That is, while harmonics matter, we believe our filtering and analysis choices have removed any way for them to appear in these data. We have added a sentence to this point in the Methods.

The limitations of the 1 hour discontinuation related to rebound are adequately acknowledged.

Thank you.

Reviewer #3 (Remarks to the Author):

This is an important and well-conceived study. It provides a valuable new framework for the design and evaluation of DBS for psychiatric diseases. An unfortunate feature of the revised report is that the authors continue to over-emphasize their claim that alterations in “cognitive control” underlie “mental illness” (MDD, OCD, . . .), even when the measure of cognitive control that they used in this study does not clearly support this. I would encourage the authors to restrict their conclusions to the circumscribed findings drawn from their study results. This issue can be further evaluated in future studies. There are some specific changes I would request, however:

P. 1: Remove “cognitive and physiologic” before “mechanisms of action”. Opening up

the idea that there are “cognitive” mechanisms of action for DBS unnecessarily enters into conceptually flawed/confused territory that is best avoided in a leading science journal.

Removed.

The authors mention “noninvasive stimulation” at various points throughout the report, including: “Non-invasive stimulation can enhance cortical oscillations if very carefully designed, and might achieve the same clinical benefits without the risks and expense of neurosurgery.” There are many psychotherapeutic, pharmacotherapeutic, and other treatment modalities that could be useful to treat psychiatric disease. This study investigated DBS. Extending the findings from this study to speculations about the possible benefits of other treatment modalities is unnecessary and unsupported.

Sentences removed on pp 4 and 14 to eliminate references to non-invasive stimulation. Your point is reasonable.